# Label-invariant Augmentation for Semi-Supervised Graph Classification

**Han Yue**    **Chunhui Zhang**    **Chuxu Zhang**    **Hongfu Liu**
Michtom School of Computer Science
Brandeis University, Waltham, MA
{hanyue,chunhuizhang,chuxuzhang,hongfuliu}@brandeis.edu

## Abstract

Recently, contrastiveness-based augmentation surges a new climax in the computer vision domain, where some operations, including rotation, crop, and flip, combined with dedicated algorithms, dramatically increase the model generalization and robustness. Following this trend, some pioneering attempts employ the similar idea to graph data. Nevertheless, unlike images, it is much more difficult to design reasonable augmentations without changing the nature of graphs. Although exciting, the current graph contrastive learning does not achieve as promising performance as visual contrastive learning. We conjecture the current performance of graph contrastive learning might be limited by the violation of the label-invariant augmentation assumption. In light of this, we propose a label-invariant augmentation for graph-structured data to address this challenge. Different from the node/edge modification and subgraph extraction, we conduct the augmentation in the representation space and generate the augmented samples in the most difficult direction while keeping the label of augmented data the same as the original samples. In the semi-supervised scenario, we demonstrate our proposed method outperforms the classical graph neural network based methods and recent graph contrastive learning on eight benchmark graph-structured data, followed by several in-depth experiments to further explore the label-invariant augmentation in several aspects.

## 1 Introduction

Contrastive augmentation aims to expand training data in both volume and diversity in a self-supervised fashion to increase model robustness and generalization. Common sense and domain knowledge are employed to design the contrastive augmentation operations. Denoising auto-encoder [2, 3] is one of the pioneering studies to apply perturbations to generate contrastive samples for tablet data, which takes a corrupted input and recovers the original undistorted input. For visual data, some operations, including rotation, crop, and flip, combined with dedicated algorithms, significantly improve the learning performance in diverse tasks [28, 8, 26, 19, 6]. Treating the augmented and original samples as positive pairs and the augmented samples from different source samples as negative pairs, contrastive learning aims to learn the augment-invariant representations by increasing the similarity of positive pairs and the dissimilarity of negative pairs [5]. These positive pairs increase the model robustness due to the assumption that the augmented operations preserve the nature of images and make the augmented samples have consistent labels with the original ones. The negative pairs work as the instance-level discrimination, which is expected to enhance the model generalization, but might deteriorate the downstream task since the negative pairs contain the augmented samples from different source samples but with the same category. The recent BYOL [16] and SimSiam [7] demonstrate the negative effect of the negative pairs and conclude that the current performance of contrastive learning can be further boosted even without negative pairs.

36th Conference on Neural Information Processing Systems (NeurIPS 2022).

Following this trend, some pioneering attempts employ contrastive augmentation to graph data [11]. GraphCL [40] is the first work to address the graph contrastive learning problem with four types of augmentations, including node dropping, edge perturbation, attribute masking, and subgraph extraction. Later, JOAO [39] extends GraphCL by automatically selecting one type of graph augmentation from the above four types plus non-augmentation. GRACE [42] treats the original graph data and the novel-level augmented data as two views and learns the graph representation by maximizing the agreement between the two views. Similarly, MVGRL [18] conducts the contrastive multi-view representation learning on both node and graph levels. Beyond the above studies to augment graphs, simGRACE [36] perturbs the model parameters for contrastive learning, which can be regarded as an ensemble of model perturbation or a robust regularization. Although exciting, the above studies point out that the effectiveness of graph contrastive learning heavily hinges on ad-hoc data augmentations, which need to be carefully designed or selected per dataset and request more domain knowledge.

**Contributions**. We conjecture these hand-crafted graph augmentations might change the nature of the original graph and violate the label-invariant assumption in the downstream tasks. Different from treating graph contrastive learning in a pre-trained perspective, we aim to incorporate the downstream classification task into the representation learning, where the label information is fully used for both decision boundary learning and graph augmentation. Specifically, we propose Graph Label-invariant Augmentation (GLA), which conducts augmentation in the representation space and augments the most difficult sample while keeping the label of the augmented sample the same as the original sample. Our major contributions are summarized as follows:

- We propose a label-invariant augmentation strategy for graph contrastive learning, which involves labels in the downstream task to guide the contrastive augmentation. It is worthy to note that we do not generate any graph data. Instead, we directly generate the label-consistent representations as the augmented graphs during the training phase.

- In the rich representation space, we aim to generate the most difficult sample for the model and increase the model generalization. We choose a lightweight technique by randomly generating a set of qualified candidates and selecting the most difficult one, i.e., minimizing the maximum loss or worst case loss over the augmented candidates.

- We conduct a series of semi-supervised experiments on eight graph benchmark datasets in a fair setting and compare our label-invariant augmentation with classical graph neural network based methods and recent graph contrastive methods by running the codes provided by the original authors. Extensive results demonstrate our label-invariant augmentation can achieve better performance in general cases without generating real augmented graphs and any specific domain knowledge. Besides algorithmic performance, we also provide rich and in-depth experiments to explore label-invariant augmentation in several aspects.

## 2 Related Work

Here we introduce the related work of graph neural networks and graph contrastive learning for graph classification. Node classification, although related, is not covered here due to its different setting.

**Graph Neural Network**. Graph Neural Networks (GNNs) have been employed on various graph learning tasks and achieved promising performance [23]. To extract the representation of each node, GNNs pass node embeddings from its connected neighbor nodes and apply feedforward neural networks to transform the aggregated features. As a pioneer study in GNNs, graph convolutional network (GCN) firstly aims to generalize the convolution mechanism from image to graph [23, 35, 14]. Based on GCN, instead of simply summing and averaging connected neighboring node's embedding, graph attention networks [31, 34, 33, 41, 13] adopt an attention mechanism that builds self-attention to score each connected neighboring nodes' embedding to identify the more important nodes and enhance the effectiveness of message passing. Then in order to break prior GNN's limitations on message passing over long distances on large graphs, graph recurrent neural networks [17, 9] apply the gating mechanism from RNNs to propagation on graph topology. Simultaneously, for dealing with the noise introduced from more than 3 layers of graph convolution, DeepGCN [25, 24] uses skip connections and enables GCN to achieve better results with deeper layers. Recently, GAE [22] and Infomax [30] achieve state-of-the-art performance on several benchmark datasets. GAE extends the variational auto-encoder to graph neural networks for unsupervised learning, while Infomax learns

the unsupervised representation on graphs to enlarge mutual information between local (node-level) and global (graph-level) representations in one graph.

**Graph Contrastive Learning**. Recently, many studies have been devoted to the graph contrastive learning area in diverse angles, including graph augmentation, negative sample selection, and view fusion. GraphCL [40] summarizes four types of graph augmentations to learn the invariant representation across different augmented views. Built on GraphCL, JOAO [39] proposes a learnable module to automatically select augmentation for different datasets to alleviate the human labor in combinations of these augmentations. Differently, MVGRL [18] contrasts node and graph encodings across views which enriches more negative samples for contrastive learning. Later, InfoGCL [37] diminishes the mutual information between contrastive parts among views while preserving the task-relevant representation. Beyond augmenting graphs, SimGRACE [36] disturbs the model weights and then learns the invariant high-level representation at the output end to alleviate the design of graph augmentation.

Different from the above methods that separate the pre-train and fine-tuning phases, we aim to employ the label information in downstream tasks to guide the augmentation process. Specifically, in this study, we propose a label-invariant augmentation strategy for graph-level representation learning.

## 3 Methodology

A graph can be represented by $G = (V, X, A)$, where $V = \{v_1, v_2, ..., v_n\}$ is the set of vertexes, $X \in \mathbb{R}^{n \times d}$ denotes the features of each vertex, and $A \in \{0, 1\}^{n \times n}$ represents the adjacency matrix. Given a set of labeled graphs $\mathcal{S} = \{(G_1, y_1), (G_2, y_2), ..., (G_M, y_M)\}$ where $M$ is the number of labeled graphs, and $y_i \in \mathcal{Y}$ is the corresponding categorical label of graph $G_i \in \mathcal{G}$ ($1 \leq i \leq M$), and another set of unlabeled graphs $\mathcal{T} = \{G_{M+1}, ..., G_N\}$, where $N$ is the number of all graphs, $M < N$, the semi-supervised graph classification problem can be defined as learning a mapping function from graphs to categorical labels $f : \mathcal{G} \rightarrow \mathcal{Y}$ to predict the labels of $\mathcal{T}$. In this section, we first illustrate our motivation supported by empirical evidence, then we elaborate on our Graph Label-invariant Augmentation (GLA) method for semi-supervised graph classification.

### 3.1 Motivation

Augmentation plays an important role in neural network training. It not only improves the robustness of learned representation but also introduces rich data for training. For graph-structured data, GraphCL [40] proposes four types of augmentations: node dropping, edge perturbation, attribute masking, and subgraph sampling. However, it is widely noticed that the effectiveness of graph contrastive learning highly depends on the chosen types of augmentations for specific graph data [39, 39]. To illus-

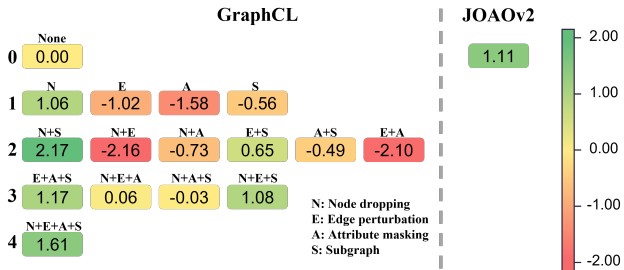

Figure 1: Performance gains (%) of GraphCL and JOAOv2 under different augmentation settings on *MUTAG* [10] dataset compared to none augmentation setting.

trate the difference between various augmentation combinations, we conduct experiments on *MUTAG* [10] in a semi-supervised graph classification task, where the label rate is set to 50%. Figure 1 shows the performance gains (classification accuracy %) of different augmentation combinations compared to none augmentation. We can see that different augmentation combinations result in different performances, and JOAOv2 automatically selects data augmentations but cannot guarantee to outperform GraphCL with all augmentation combinations. Moreover, some augmentation combinations work worse than none augmentation, which demonstrates that some augmentations hurt the model training. We further fine-tune a model with 100% labeled graphs from *MUTAG*, and then feed the augmented graphs (randomly selected from the four augmentation types with an augmentation ratio of 0.2) to this model, finding that about 80% augmented graphs have the same labels with their corresponding original graphs. It indicates that most augmentations are reasonable, which is one of the reasons that GraphCL works well. While on the other hand, there are still about 20% augmented

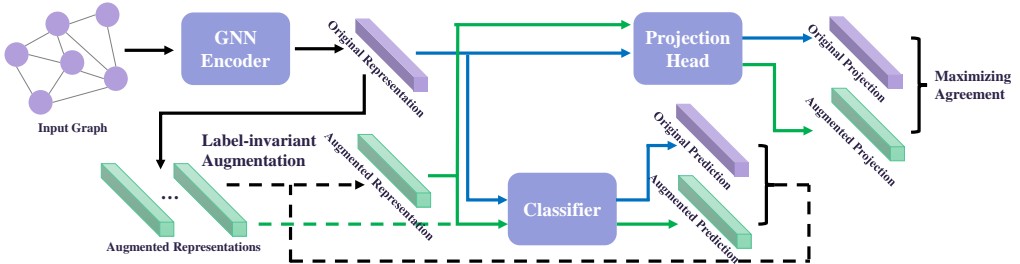

Figure 2: Framework of our Graph Label-invariant Augmentation (GLA) for semi-supervised graph classification. Given an input graph, a Graph Neural Network (GNN) Encoder is employed to encode the input graph into a graph-level representation (original representation). Then we perturb the original representation to get multiple augmented representations. A classifier is adopted to verify whether the augmentations are label-invariant or not. We select the "hardest" augmented representation, i.e., the one that has the least probability of belonging to the same class as ground-truth label/original representation for labeled/unlabeled graph, from all augmented representations that satisfy the label-invariant constraint. On top of GNN Encoder, we build a projection head to get projections for both original representation and label-invariant augmented representation. We maximize the agreement between projections via a contrastive loss for all graphs and refine the classifier via a cross-entropy loss with labeled graphs.

graphs getting different labels from the original ones. Motivated by this, we design a label-invariant augmentation strategy for graph contrastive learning.

## 3.2   Label-invariant Augmentation

**Framework Overview**. Figure 2 illustrates the framework of our proposed Graph Label-invariant Augmentation (GLA) for semi-supervised graph classification, which mainly consists of four components: Graph Neural Network Encoder, Classifier, Label-invariant Augmentation, and Projection Head. We first use GNN Encoder to get graph-level original representation for the input graph. Then Label-invariant Augmentation, together with Classifier, is utilized to generate augmented representation from original representation under a label-invariant constraint. For an unlabeled graph, we expect that the labels represented by original prediction and augmented prediction are the same. For a labeled graph, we expect that the label represented by augmented prediction is the same as the ground truth label. A cross-entropy loss is used to keep refining the classifier with labeled graphs. Finally, a Projection Head is adopted to generate projections for contrastive loss. We use $\Theta = \{\theta_G, \theta_C, \theta_P\}$ to denote the trainable parameters set, where $\theta_G$, $\theta_C$, and $\theta_P$ denote the parameters of GNN Encoder, Classifier and Projection Head, respectively. Details of each component are as follows.

**Graph Neural Network Encoder**. Graph Neural Network Encoder aims to get graph-level representations for graph-structured data. It is flexible to adopt various GNNs for this part. We follow GraphCL [40] and utilize ResGCN [4], which takes Graph Convolutional Network (GCN) [21] as the backbone, to extract node-level representations from the input graph, and then a global sum pooling layer is used to obtain its graph-level representation. The computation of the GCN layer with the parameter $\theta_G$ is described as follows:

$$G^{(l+1)} = \sigma(\tilde{D}^{-\frac{1}{2}}\tilde{A}\tilde{D}^{-\frac{1}{2}}G^{(l)}\theta_G^{(l)}), \qquad (1)$$

where $\tilde{A} = A + I_n$ is the adjacency matrix $A$ with added self-connections, $I_n \in \mathbb{R}^{n \times n}$ is the identity matrix, $\tilde{D}$ is the degree matrix of $\tilde{A}$, and $\theta_G^{(l)}$ is a layer-specific trainable weight matrix. $G^{(l)}$ denotes the matrix in the $l$-th layer, and $G^{(0)} = X$. We employ $\sigma(\cdot) = \text{ReLU}(\cdot)$ as the activation function.

Then on top of the ResGCN, we use a global sum pooling layer to get graph-level representations from node-level representations as follows:

$$H = \text{Pooling}(G). \qquad (2)$$

Here we use $H^O$ to denote the original representation of the input graph and $H^A$ to denote the augmented representation of the augmented graph. The augmentation method will be described in the Label-invariant Augmentation part.

**Classifier**. Based on the graph-level representations, we employ fully-connected layers with the parameter $\theta_C$ for prediction:

$$C^{(l+1)} = \sigma(C^{(l)} \cdot \theta_C^{(l)}), \tag{3}$$

where $C^{(l)}$ denotes the embeddings in the $l$-th layer, and the input layer $C^{(0)} = H^O$ or $C^{(0)} = H^A$ for the original representation and augmented representation, respectively. In our experiments, we adopt a 2-layer multilayer perceptron and obtain predictions $C^O$ and $C^A$ for original representation $H^O$ and augmented representation $H^A$, and $\sigma(\cdot) = \mathrm{ReLU}(\cdot)$ for the first layer and $\sigma(\cdot) = \mathrm{Softmax}(\cdot)$ for the second layer as the activation function.

**Label-invariant Augmentation**. Instead of augmenting graph data by node dropping, edge perturbation, attribute masking, or subgraph sampling as recent graph contrastive learning methods [40, 39], we conduct the augmentation in the representation space by adding a perturbation to the original representation $H^O$ so that we do not need to generate any graph data. In our experiment, we first calculate the centroid of original representations for all graphs and get the average value of euclidean distances between each original representation and the centroid as $d$, that is:

$$d = \frac{1}{N} \sum_{i=1}^{N} \| H_i^O - \frac{1}{N} \sum_{j=1}^{N} H_j^O \|. \tag{4}$$

Then the augmented representation $H^A$ is calculated by:

$$H^A = H^O + \eta d\Delta, \tag{5}$$

where $\eta$ scales the magnitude of the perturbation, and $\Delta$ is a random unit vector.

Based on the classifier formulated in Eq. (3), we define label-invariant as follows. For labeled graphs, label-invariant means the predictions of augmented representations by the classifier are the same as their corresponding ground-truth labels. For unlabeled graphs, label-invariant denotes that the predictions of augmented representations and the predictions of original representations by the classifier are the same.

To achieve the label-invariant augmentation, for each graph, we randomly generate multiple perturbations and select the qualified augmentation candidates that obey the label-invariant property. Among these qualified candidates, we choose the most difficult one, i.e., the one that has the least probability of belonging to the same class as ground-truth label/original representation for labeled/unlabeled graph, to increase the model generalization ability.

**Projection Head**. We employ fully-connected layers with the parameter $\theta_P$ to get projections for contrastive learning from graph-level representations, which is shown as:

$$P^{(l+1)} = \sigma(P^{(l)} \cdot \theta_P^{(l)}). \tag{6}$$

We adopt a 2-layer multilayer perceptron and get projections $P^O$ and $P^A$ from original representation $H^O$ and augmented representation $H^A$.

**Objective Function**. Our objective function consists of contrastive loss and classification loss. For contrastive loss, we utilize the normalized temperature-scaled cross-entropy loss (NT-Xent) [40] but only keep the positive-pair part as follows:

$$\mathcal{L}_P = \frac{-(P^O)^\top P^A}{\|P^O\|\|P^A\|}. \tag{7}$$

Maximizing the agreement between original projection and augmented projection would increase the robustness of the model.

For classification loss, we adopt cross-entropy, which is defined as:

$$\mathcal{L}_C = - \sum_{i=1}^{c} (Y_i^O \log P_i^O + Y_i^O \log P_i^A), \tag{8}$$

where $Y^O$ is the label of the input graph, and $c$ is the number of graph categories. We only calculate $\mathcal{L}_C$ for labeled graphs. The improvement of the classifier would help with the label-invariant augmentation, which in turn benefits the training of the classifier.

Table 1: Statistics of datasets for semi-supervised graph classification

| Datasets | Category | #Class | #Graph | Avg. #Node | Avg. #Edge |
|----------|----------|--------|--------|------------|------------|
| *MUTAG* | Biochemical Molecules | 2 | 188 | 17.93 | 19.79 |
| *PROTEINS* | Biochemical Molecules | 2 | 1,113 | 39.06 | 72.82 |
| *DD* | Biochemical Molecules | 2 | 1,178 | 284.32 | 715.66 |
| *NCI1* | Biochemical Molecules | 2 | 4,110 | 29.87 | 32.30 |
| *COLLAB* | Social Networks | 3 | 5,000 | 74.49 | 2,457.78 |
| *RDT-B* | Social Networks | 2 | 2,000 | 429.63 | 497.75 |
| *RDT-M5K* | Social Networks | 5 | 4,999 | 508.52 | 594.87 |
| *GITHUB* | Social Networks | 2 | 12,725 | 113.79 | 234.64 |

Combining Eq. (7) and (8), our overall objective function can be written as follows:

$$\min_{\Theta} \mathcal{L}_P + \alpha \mathcal{L}_C, \tag{9}$$

where $\alpha$ is a trade-off hyperparameter to balance the contrastive loss and classification loss.

**Discussion**. From the perspective of information usage for model training, our proposed method is the same as the semi-supervised learning task by recent graph contrastive learning methods [40, 39, 18, 36], which use structure information of all graphs and label information of a subset of all graphs for model training. From the perspective of training strategy, the previous methods first pre-train a model via a contrastive loss and then fine-tune the model for downstream tasks. While our proposed method focuses on semi-supervised classification, we merge the pre-train and fine-tuning phases into one integrated phase. During our training phase, the augmented samples increase the model robustness and generalization ability, and the classifier helps to generate better augmented samples, which in turn benefits classification performance. Theoretically, our method is well supported by MaxUp [15], where the hardest sample augmentation can be regarded as a gradient-norm regularization.

# 4 Experiments

In this section, we first describe our semi-supervised settings of experiments and baseline methods for comparison. Then we show the algorithmic performance of these methods on eight graph benchmark datasets in a fair setting. Finally, we provide some insightful experiments to demonstrate the effectiveness of the proposed Graph Label-invariant Augmentation (GLA) method.

## 4.1 Experiment Settings

**Datasets**. We select eight public graph classification benchmark datasets from TUDataset [27] for evaluation, including *MUTAG* [10], *PROTEINS* [1], *DD* [12], *NCI1* [32], *COLLAB* [38], *RDT-B* [38], *RDT-M5K* [38], and *GITHUB* [29]. Table 1 shows the statistics of these datasets. The first four datasets include biochemical molecules and proteins, and the last four datasets are about social networks. The numbers of graphs in these datasets range from 188 to 12,725, the average node numbers range from 17.93 to 508.52, and the average edge numbers are from 19.79 to 2,457.78, indicating the diversity of these datasets.

**Compared Methods and Implementation**. We choose two heuristic self-supervised methods, GAE [22] and Infomax [30], and four recent graph contrastive learning methods, MVGRL [18], GraphCL [40], JOAOv2 [39], and SimGRACE [36], for comparison on semi-supervised graph classification task. GAE performs adjacency matrix reconstruction by using a graph convolutional network (GCN) [21] encoder and a simple inner product decoder. Infomax is based on global-local representation consistency enforcement, which maximizes the mutual information between global and local representation. MVGRL proposes to learn node and graph level representations by node diffusion and contrasting encodings. GraphCL presents four types of graph augmentations. Based on GraphCL, JOAOv2 is designed as a unified bi-level optimization framework to automatically select graph augmentations. SimGRACE perturbs parameters of graph encoder for contrastive learning, which does not require data augmentations.

For GAE, Infomax, and GraphCL, we adopt the implementations and default hyperparameter settings provided by the source codes of GraphCL [40]. For other compared methods, we follow the implementations and hyperparameter settings in their corresponding source codes. The compared methods are pre-trained first and then fine-tuned for the semi-supervised graph classification task. For

Table 2: Semi-supervised graph classification results (Accuracy % ± Standard Deviation %) on eight benchmark datasets. The best and second-best results are highlighted in red and blue, respectively.

| Label | Methods | MUTAG | PROTEINS | DD | NCI1 | COLLAB | RDT-B | RDT-M5K | GITHUB | Avg. | Rank |
|---|---|---|---|---|---|---|---|---|---|---|---|
| 30% | GAE | 83.63 ± 0.81 | 74.31 ± 0.33 | 77.33 ± 0.36 | 77.20 ± 0.22 | 77.46 ± 0.11 | 90.75 ± 0.17 | 54.81 ± 0.18 | 65.22 ± 0.11 | 75.09 | 5.00 |
| | Infomax | 84.68 ± 1.12 | 74.84 ± 0.28 | 77.07 ± 0.45 | 79.49 ± 0.17 | 77.30 ± 0.19 | 90.65 ± 0.17 | 55.37 ± 0.20 | 66.45 ± 0.06 | 75.73 | 4.25 |
| | MVGRL | 83.16 ± 0.98 | 75.56 ± 0.44 | 77.08 ± 0.56 | 72.41 ± 0.18 | 75.28 ± 0.12 | 88.20 ± 0.16 | 53.16 ± 0.06 | 64.71 ± 0.04 | 73.70 | 6.12 |
| | SimGRACE | 83.68 ± 0.84 | 74.38 ± 0.30 | 76.27 ± 0.38 | 78.52 ± 0.17 | 78.66 ± 0.24 | 90.60 ± 0.17 | 55.54 ± 0.16 | 66.81 ± 0.14 | 75.56 | 4.50 |
| | GraphCL | 85.20 ± 0.98 | 74.12 ± 0.30 | 78.60 ± 0.37 | 79.22 ± 0.09 | 77.90 ± 0.20 | 90.35 ± 0.18 | 56.07 ± 0.15 | 67.63 ± 0.13 | 76.14 | 3.38 |
| | JOAOv2 | 85.67 ± 0.91 | 75.02 ± 0.30 | 77.16 ± 0.30 | 78.69 ± 0.18 | 79.88 ± 0.17 | 91.65 ± 0.15 | 55.23 ± 0.14 | 67.96 ± 0.10 | 76.41 | 2.62 |
| | GLA (Ours) | 86.32 ± 1.25 | 75.65 ± 0.37 | 77.49 ± 0.40 | 79.71 ± 0.13 | 78.78 ± 0.12 | 91.05 ± 0.25 | 55.85 ± 0.22 | 65.16 ± 0.19 | 76.25 | 2.12 |
| 50% | GAE | 84.12 ± 0.90 | 74.75 ± 0.38 | 78.35 ± 0.31 | 79.56 ± 0.16 | 80.47 ± 0.14 | 90.95 ± 0.19 | 55.69 ± 0.16 | 67.09 ± 0.13 | 76.37 | 5.88 |
| | Infomax | 87.37 ± 1.11 | 75.38 ± 0.38 | 78.26 ± 0.38 | 80.80 ± 0.13 | 79.70 ± 0.11 | 91.50 ± 0.26 | 56.51 ± 0.18 | 67.70 ± 0.09 | 77.15 | 3.75 |
| | MVGRL | 85.79 ± 0.23 | 76.72 ± 0.34 | 78.60 ± 0.46 | 74.09 ± 0.10 | 76.08 ± 0.05 | 88.55 ± 0.06 | 54.04 ± 0.06 | 64.89 ± 0.05 | 74.84 | 5.62 |
| | SimGRACE | 86.32 ± 0.88 | 75.09 ± 0.35 | 78.39 ± 0.35 | 79.78 ± 0.24 | 80.48 ± 0.15 | 91.45 ± 0.16 | 56.50 ± 0.20 | 67.71 ± 0.16 | 76.97 | 4.50 |
| | GraphCL | 87.28 ± 0.71 | 75.29 ± 0.29 | 78.73 ± 0.46 | 80.17 ± 0.19 | 80.40 ± 0.16 | 91.45 ± 0.25 | 56.83 ± 0.19 | 68.71 ± 0.09 | 77.36 | 3.25 |
| | JOAOv2 | 86.78 ± 0.79 | 75.74 ± 0.29 | 78.52 ± 0.45 | 80.10 ± 0.17 | 81.50 ± 0.18 | 92.10 ± 0.16 | 56.51 ± 0.17 | 68.97 ± 0.11 | 77.53 | 2.75 |
| | GLA (Ours) | 90.00 ± 0.94 | 76.19 ± 0.28 | 80.22 ± 0.37 | 80.66 ± 0.28 | 80.84 ± 0.12 | 91.65 ± 0.22 | 56.63 ± 0.13 | 66.59 ± 0.14 | 77.85 | 2.25 |
| 70% | GAE | 87.31 ± 0.66 | 75.47 ± 0.38 | 79.37 ± 0.36 | 79.78 ± 0.17 | 80.78 ± 0.12 | 91.50 ± 0.19 | 56.25 ± 0.16 | 68.42 ± 0.14 | 77.36 | 5.62 |
| | Infomax | 88.33 ± 0.73 | 75.92 ± 0.38 | 79.28 ± 0.33 | 82.85 ± 0.16 | 81.04 ± 0.12 | 92.15 ± 0.13 | 56.63 ± 0.18 | 68.88 ± 0.14 | 78.14 | 3.62 |
| | MVGRL | 87.95 ± 0.35 | 77.81 ± 0.35 | 79.51 ± 0.34 | 74.43 ± 0.08 | 76.42 ± 0.08 | 88.65 ± 0.23 | 54.40 ± 0.11 | 65.00 ± 0.08 | 75.52 | 5.25 |
| | SimGRACE | 87.37 ± 0.71 | 76.52 ± 0.36 | 78.90 ± 0.29 | 81.80 ± 0.15 | 81.88 ± 0.23 | 92.45 ± 0.13 | 56.58 ± 0.09 | 68.19 ± 0.15 | 77.96 | 4.12 |
| | GraphCL | 88.33 ± 0.86 | 76.36 ± 0.25 | 79.03 ± 0.29 | 82.50 ± 0.13 | 81.08 ± 0.17 | 91.85 ± 0.14 | 56.91 ± 0.17 | 69.19 ± 0.08 | 78.16 | 3.62 |
| | JOAOv2 | 87.78 ± 0.76 | 76.46 ± 0.27 | 79.11 ± 0.38 | 81.70 ± 0.26 | 82.16 ± 0.17 | 92.20 ± 0.19 | 56.67 ± 0.16 | 69.96 ± 0.11 | 78.26 | 3.25 |
| | GLA (Ours) | 91.05 ± 0.86 | 77.45 ± 0.38 | 80.71 ± 0.29 | 83.24 ± 0.14 | 81.54 ± 0.14 | 91.70 ± 0.17 | 57.01 ± 0.14 | 67.11 ± 0.18 | 78.73 | 2.50 |

(a) performance gain    (b) label-invariant rate distribution    (c) GLA's label-invariant rate

Figure 3: Performance gain and label-invariant rates. (a) demonstrates the average performance gains on eight datasets with more labeled samples produced by GraphCL, JOAOv2, and GLA. (b) shows the label-invariant rate distributions of different augmentation methods over eight datasets. (c) shows the label-invariant rates of our GLA over different semi-supervised settings.

our proposed Graph Label-invariant Augmentation (GLA) method,[1] we perform contrastive learning and graph classifier learning synchronously. The implementation details of GLA are as follows. We implement the networks based on GraphCL [40] by PyTorch, set the magnitude of perturbation $\eta$ to 1.0, and the weight of classification loss $\alpha$ to 1.0, which is the same with GraphCL. We adopt Adam optimizer [20] to minimize the objective function in Eq. (9).

**Evaluation Protocol**. We evaluate the models with 10-fold cross-validation. We randomly shuffle a dataset and then evenly split it into 10 parts. Each fold corresponds to one part of data as the test set and another part as the validation set to select the best epoch, where the rest folds are used for training. We select 30%, 50%, 70% graphs from the training set as labeled graphs for each fold, then conduct semi-supervised learning. For a fair comparison, we use the same training/validation/test splits for all compared methods on each dataset, and report the average accuracy across 10 folds.

## 4.2 Algorithmic Performance

Table 2 shows the prediction results of two self-supervised and five graph contrastive learning methods under the semi-supervised graph classification setting with 30%, 50%, and 70% label ratios on eight benchmark datasets, where the best and second-best results are highlighted in red and blue, respectively, and the last column is the average rank score across all datasets. Although different algorithms achieve their best performances on different datasets, the contrastiveness-based methods perform better than the non-contrastiveness-based methods in general, which indicates the effectiveness of the graph augmentation. Our proposed GLA achieves the best ranking scores under all 30%, 50%, and 70% label ratios in experiments, the second-best average performance

---

[1]Our code is available at `https://github.com/brandeis-machine-learning/GLA`

under 30% label ratio, and the best average performance under 50% and 70% label ratios. In our algorithmic design, we employ the decision boundary learned from the labeled samples to verify the label-invariant augmentation. It is worthy to note that the quality of the decision boundary depends on the number of labeled samples. We conjecture that a 30% label ratio is not sufficient enough to learn a high-quality decision boundary, resulting in our GLA performing slightly worse than JOAOv2 on average. With more labeled samples, our GLA delivers the best average performance over other competitive methods. Different from other graph contrastive learning methods, our augmentation method aims to generate label-invariant augmentations, which decreases the possibility of getting "bad" augmentations, thus resulting in better performance.

Besides the general comparison in Table 2, we dive into details and discover several interesting findings. Figure 3(a) demonstrates the average performance gains on eight datasets with more labeled samples produced by GraphCL, JOAOv2, and our GLA, the top three methods in our experiments. In addition to seeing that the increased performance of all three methods well aligns with more labeled samples, our GLA receives more performance gains than GraphCL and JOAOv2. By such comparisons, we can roughly eliminate the effect of more labeled samples and attribute the extra gains to the label-invariant augmentation. It also verifies our aforementioned conjecture that the high-quality decision boundary is beneficial to the label-invariant augmentation, further bringing in the performance boost. Moreover, we further verify our motivation by checking the label-invariant property of different contrastive methods. While we do not have a ground truth classifier, we use fine-tuned classifiers in the representation spaces learned by these contrastive methods with a 100% label ratio as the surrogates of the ground truth classifier. Then we use these classifiers to assess how many of the augmented representations belong to the same class as their corresponding original representations. Figure 3(b) presents the distributions of label-invariant rates across eight baseline datasets for all graph contrastive methods. As our GLA trained under different label ratios would generate different augmentations, we put the results of GLA's label-invariant rates under 30%, 50%, and 70% label ratios together for plotting. We can see that GLA has the highest label-invariant rates on average compared to other methods. It is also noticed that the label-invariant rates of different contrastive methods keep the same ranking with the performance in Table 2 (the last column), which verifies our motivation for designing a label-invariant augmentation strategy. Moreover, we further demonstrate our GLA's label-invariant rates along with different label ratios in Figure 3(c), which accords with our expectation that more labeled samples lead to a high-quality decision boundary and further promote the label-invariant rate in GLA.

### 4.3   In-depth Exploration

We further explore GLA in terms of negative pairs, augmentation space, and strategy.

*Negative Pairs*. The existing graph contrastive learning methods treat the augmented graphs from different source samples as negative pairs and employ the instance-level discrimination on these negative pairs. Since these methods separate the pre-train and fine-tuning phases, the negative pairs contain the augmented samples from different source samples but with the same category in the downstream tasks. Here we explore the effect of negative pairs on our GLA. Figure 4(a) shows the performance of our GLA with and without negative pairs on four datasets. We can see the performance with negative pairs significantly drops compared with our default setting without negative pairs, which behaves consistently on all four datasets. Different from the existing graph contrastive methods, our GLA integrates the pre-train and fine-tuning phases, where the negative pairs designed in a self-supervised fashion are not beneficial to the downstream tasks. This finding is also in accord with the recent studies [7, 16] in the visual contrastive learning area.

*Augmentation Space*. Different from the most graph contrastive learning methods that directly augment raw graphs, our GLA conducts the augmentation in the representation space, as we believe the raw graphs can be mapped into the representation space, and this space is much easier to augment than the original graph space. In Eq. (5), we design our representation augmentation with a random unit vector scaled by the magnitude of the perturbation $\eta$. Figure 4(b,c,e,f) show the performance of our GLA with different values of $\eta$ on four datasets, where we provide GraphCL and GraphCL+Label-Invariant as references. GraphCL+Label-Invariant takes the augmented graph from GraphCL and filters the augmented samples that violate the label-invariant property by the downstream classifier. Comparing the two references, we can see that the label-invariant property benefits not only our GLA but also other contrastive methods in most cases. For our GLA, although the $\eta$ values corresponding

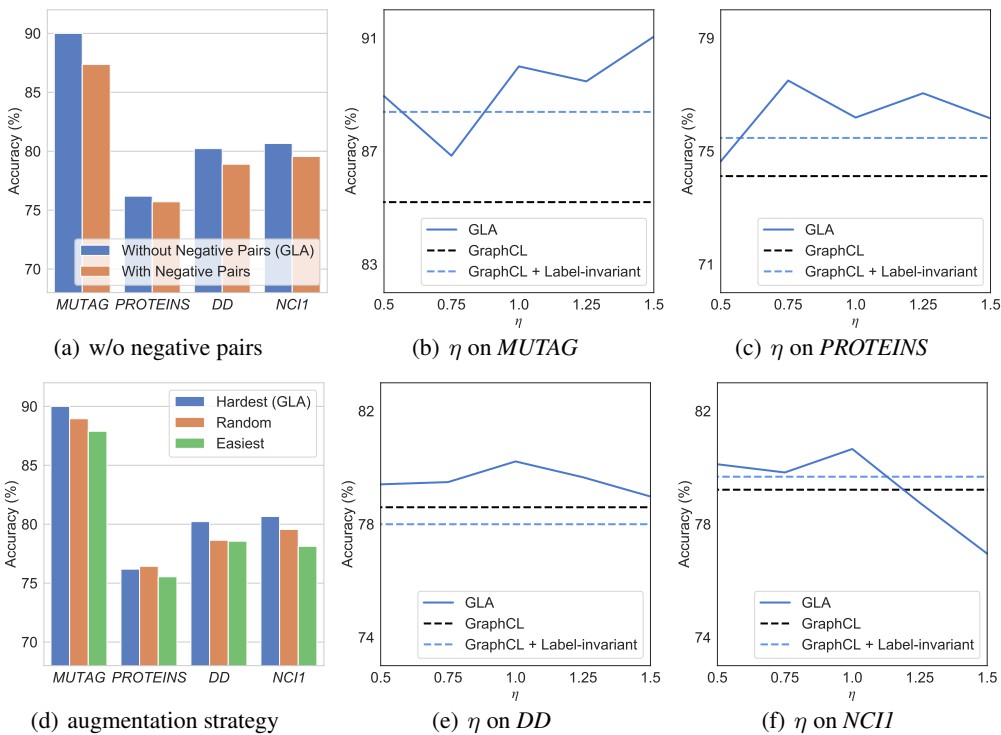

Figure 4: In-depth exploration of GLA. (a) contrastive loss with/without negative pairs, (d) performance of different label-invariant augmentation strategies, (b,c,e,f) performance of magnitude of perturbation $\eta$ on different datasets under 50% label ratio.

to the best performance vary on different datasets, the default setting with $\eta = 1$ delivers satisfying performance in general, which outperforms GraphCL+Label-Invariant and indicates the superior of the representation augmentation over the raw graph augmentation. *Augmentation Strategy.* In the representation space, there might exist multiple qualified candidates that obey the label-invariant property. Our GLA chooses the most difficult augmentation for the model. Here we demonstrate the performance of different augmentation strategies among qualified candidates, including the most difficult augmentation, random augmentation, and the easiest augmentation in Figure 4(d), where the random augmentation can be regarded as GraphCL+Label-Invariant. We can see that the most difficult augmentation increases the model generalization and indeed brings in significant improvements over the other two ways. This also provides good support for our representation augmentation, where we can find the most difficult augmentation in the representation space, but it is difficult to directly generate the raw graphs that are challenging to the downstream classifier.

## 5 Conclusion

In this paper, we consider the graph contrastive learning problem. Different from the existing methods from the pre-train perspective, we propose a novel Graph Label-invariant Augmentation (GLA) algorithm which integrates the pre-train and fine-tuning phases to conduct the label-invariant augmentation in the representation space by perturbations. Specifically, GLA first checks whether the augmented representation obeys the label-invariant property and chooses the most difficult sample from the qualified samples. By this means, GLA achieves the contrastive augmentation without generating any raw graphs and also increases the model generalization. Extensive experiments in the semi-supervised setting on eight benchmark graph datasets demonstrate the effectiveness of our GLA. Moreover, we also provide extra experiments to verify our motivation and explore the in-depth factors of GLA in the effect of negative pairs, augmentation space, and strategy. *Limitations*. The performance of our method relies on the quality of the decision boundary indicated by the downstream classifier. Therefore, our method requires graph label information from downstream tasks to help with model training. *Potential Negative Societal Impacts*. The problem addressed in this paper is well-defined and the experiments are based on public datasets. As far as we can see, it does not involve societal issues.

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
