# OpenReview forum: "Label-invariant Augmentation for Semi-Supervised Graph Classification"
_NeurIPS.cc/2022/Conference — NeurIPS 2022 Accept_

### Official Review · Reviewer_1vzF · 2022-06-17

**Rating:** 6
**Confidence:** 4
**Soundness:** 2 fair
**Presentation:** 3 good
**Contribution:** 3 good

**Summary:**

This paper proposes a label-invariant augmentation technique for graph-structured data. Different from the conventional node/edge modification and subgraph extraction, the authors conduct the augmentation in the representation space and generate the augmented samples in the most difficult direction while keeping the label of augmented data the same as the original samples. The experimental results demonstrate the effectiveness of the proposed augmentation strategy.

**Questions:**

1. In fact, many existing graph contrastive learning methods are also applicable to node classification on one graph. Can the proposed method be generalized to such problem?
2. The authors claim that they developed label-invariant augmentation strategy for graph contrastive learning which involves labels in the downstream task. I want to know the difference/relationship of this problem setting with supervised contrastive learning. Note that the label information can also be employed in supervised contrastive learning which does not involve the downstream classification task.
3. To present the motivation in Section 3.1, the authors use Fig. 1 with label rate being 50% to show that some combinations of existing augmentation strategies may even hurt the performance. I'm a bit concerned with the label rate 50%, as this is too high for semi-supervised learning. Maybe, if the label rate is low (e.g., 5%-10%), the advantage of some augmentations will become prominent.
4. For the part of "Label-invariant Augmentation" on Page 5, I do not think d defined in Eq. 4 represents direction. Note that this is simply the distance between the representation of a graph to the centroid (like ||a-b||^2), rather than direction vector like a-b. The \delta is simply a random vector, which is also not a direction vector. Therefore, I cannot fully understand the authors' claim that they "augment samples in most difficult direction while keeping the label of augmented data the same as the original samples." Moreover, how to keep the label of augmented data the same as the original samples?? I also did not find specific explanations on this.
5. In experiments, the authors use 30%, 50% and 70% label rates, which I also think is too high. Note that the definition of semi-supervised learning is that we only have a handful of labeled data in addition to a large amount of unlabeled data.

**Limitations:**

1. The model has some mathematical errors.
2. The label rates adopted in this paper are too high for semi-supervised learning.

**Strengths And Weaknesses:**

Pros:
1. I think the high-level idea is interesting and reasonable.
2. The paper is generally written in a clear way.
3. The empirical studies are quite convincing.


Cons:
1. It would be better if some theoretical evidence can be provided to justify the correctness of the proposed method.
2. Some details of model need further explanations.
3. It seems that the proposed method has some technical issues.

---

> ### Author Response · Authors · 2022-08-02
> **Response to Reviewer 1vzF (1/2)**
>
> Thank you for dedicating your time and energy to reviewing our paper. We would like to provide point-to-point responses as follows.
>
> **Q0. Theoretical Evidence**. Thanks for this excellent comment! Actually, our method is well supported by MaxUp [1], where the hardest sample augmentation can be regarded as a gradient-norm regularization. The difference between work and MaxUp lies in that our work focuses on the semi-supervised scenario, while MaxUp does not involve any label information for theoretical analysis. We will follow Reviewer 1vzF's suggestion, and add a description in the discussion part of Section 3.2 as follows:
>
> *Theoretically, our method is well supported by MaxUp, where the hardest sample augmentation can be regarded as a gradient-norm regularization.*
>
> >[1] Gong, Chengyue, et al. "Maxup: Lightweight adversarial training with data augmentation improves neural network training." *CVPR*. 2021.
>
> **Q1. Node Classification**. We believe this comment is an open question, rather than a weakness. We agree with Reviewer 1vzF that our method can also be applied in an existing framework for node classification task without much modification. Here we compare our method with GraphCL with public splits on *Cora* (5.17\% label rate) and *Citeseer* (3.61\% label rate) in the semi-supervised setting.
>
> Table Q1: Node classification results of our proposed method with public splits on *Cora* and *Citeseer*.
> | Method | *Cora* | *Citeseer* |
> |---|:---:|:---:|
> | GraphCL | 82.45 | 72.02 |
> | GLA (Ours) | 82.58 | 72.61 |
>
> Unlike graph classification where relationships between graphs are not given, node classification problem provides linking information between nodes. Many methods focus on utilizing this property to tackle the node classification problem. Here we focus on providing a label-invariant augmentation strategy, which can be easily incorporated into other node classification frameworks.
>
> **Q2. Compared with Supervised Contrastive  Learning**. We agree with Reviewer 1vzF that supervised contrastive learning can be tailored to handle the semi-supervised task. The difference lies in that the input of our method and other competitive methods is both the labeled and unlabeled samples, while supervised contrastive learning only takes the labeled samples as input.
>
> Here we report the performance of supervised learning, supervised contrastive learning, and semi-supervised contrastive learning by our method for comparison. Supervised contrastive learning outperforms supervised learning on all 4 datasets, indicating that label-invariant augmentation increases the model robustness and generalization ability. Semi-supervised also performs better than supervised learning, which shows the impact of unlabeled graphs.
>
> Table Q2: Supervised/Semi-supervised graph classification results of our proposed method with 50\% label rate.
> | Method | *MUTAG* | *PROTEINS* | *DD* | *NCI1* |
> |---|:---:|:---:|:---:|:---:|
> | Supervised; 50\% labeled; No   Augmentation | 86.84 | 74.28 | 77.46 | 79.05 |
> | Supervised; 50\% labeled | 87.89 | 75.80 | 79.58 | 79.61 |
> | GLA (Semi-supervised; 50\% labeled +   50\% unlabeled) | 90.00 | 76.19 | 80.22 | 80.66 |
>
> **Q3. Performance With 5\% Labeled Samples**. Thanks for this suggestion. To our best knowledge, there is no clear definition of the label rates in semi-supervised learning. But we agree with Reviewer 1vzF that if the label rate is low, the advantage of some augmentations will become prominent. Here we would like to follow the Reviewer 1vzF's comment to report the performance with 5\% labeled samples on four datasets. The results are reported by the average accuracy across 10 folds. As our computational resources cannot support finishing experiments on all 8 datasets within the limited rebuttal time, we report the first 4 datasets instead.
>
> Table Q3: Semi-supervised graph classification results with 5\% label rate on four benchmark datasets.
> | Method | *MUTAG* | *PROTEINS* | *DD* | *NCI1* | Avg. | Rank |
> |---|:---:|:---:|:---:|:---:|:---:|:---:|
> | GAE | 77.00 | 68.39 | 72.58 | 70.90 | 72.22 | 4.25 |
> | Infomax | 75.79 | 68.21 | 71.95 | 70.75 | 71.68 | 5.5 |
> | MVGRL | 71.89 | 66.39 | 69.95 | 66.80 | 68.76 | 7 |
> | SimGRACE | 81.58 | 69.46 | 72.29 | 71.17 | 73.63 | 3 |
> | GraphCL | 74.74 | 68.93 | 73.98 | 71.56 | 72.30 | 3 |
> | JOAOv2 | 74.74 | 71.07 | 72.86 | 71.02 | 72.42 | 3.25 |
> | GLA (Ours) | 82.19 | 69.10 | 73.85 | 71.82 | 74.24 | 1.75 |

---

> > ### Author Response · Authors · 2022-08-02
> > **Response to Reviewer 1vzF (2/2)**
> >
> > **Q4. Confusion on "Hardest Sample."** For the part of "Label-invariant Augmentation" on Page 5, we agree that $d$ defined in Eq. (4) is a distance instead of a direction. However, $\Delta$ is a random vector indicating a random direction (for example, in 2D space, (1,0) is a direction vector pointing along X-axis). As described in Eq. (5), an  augmented representation candidate is generated by $H^{A} = H^{O} + \eta d \Delta$, where $\eta$ and $d$ control the magnitude of the perturbation, and $\Delta$ is a random direction. In this way, we generate multiple candidates. Then based on the classifier in our model, we achieve label-invariant by keeping the label of augmented data the same as ground-truth label/original representation for labeled/unlabeled graph. Among all candidates that satisfy the label-invariant constraint, we select the one that has the least probability of belonging to the same class as ground-truth label/original representation for labeled/unlabeled graph as the hardest sample. To better explain this, we add more interpretation on this in the "Label-invariant Augmentation" part of Section 3.2 in revision as follows:
> >
> > *Based on the classifier formulated in Eq. (3), we define label-invariant as follows. For labeled graphs, label-invariant means the predictions of augmented representations by the classifier are the same as their corresponding ground-truth labels. For unlabeled graphs, label-invariant denotes that the predictions of augmented representations and the predictions of original representations by the classifier are the same.*
> >
> > *To achieve the label-invariant augmentation, for each graph, we randomly generate multiple perturbations and select the qualified augmentation candidates that obey the label-invariant property. Among these qualified candidates, we choose the most difficult one, i.e., the one that has the least probability of belonging to the same class as ground-truth label/original representation for labeled/unlabeled graph, to increase the model generalization ability.*
> >
> > **Q5. Too High Label Rates**. Please refer to Q3.

---

> ### Comment · Area_Chair_sM4g · 2022-08-08
> **To reviewers: please provide feedback to authors' response**
>
> We are approaching the end of the rebuttal period, and the authors have provided feedback to which your further response is appreciated, especially seeing there are dispute on the rating among the reviewers.

---

### Official Review · Reviewer_3fFh · 2022-07-09

**Rating:** 7
**Confidence:** 3
**Soundness:** 3 good
**Presentation:** 3 good
**Contribution:** 3 good

**Summary:**

The authors propose a graph contrastive learning approach in which the augmentations are done in feature space such that they do not change the predicted label. They present experiments on  8 different benchmarks, and show competitive results.

**Questions:**

I would like to see the questions in my first "weakness" comment adressed and explained

**Strengths And Weaknesses:**

**Strength**:
the idea is clever and well-motivated
the experiments are extensive and rather convincing
the article is clear and well written, even if some details are sometimes missing and could be more rigorous

**Weaknesses**:
- Some parts of the paper are unclear:
    - Since picking positive examples requires the labels (to ensure label invariance and "difficulty"), how are the unlabelled graph handled?
    - it is unclear how " closest to the decision boundary of the classifier" exactly translates here
- The authors keep referring to their "hard positive mining" strategy as "finding the hardest example" or "the hardest direction", but they simply pick a hard example within a sampled set, which is very standard and do not need to be presented as an approximate bilevel optimization scheme.

**Overall**:
An interesting and well-executed idea. The article is sometimes not as clear as it could be, hiding actual crucial information. I would be inclined to increase my rating if the authors could answer my doubts convincingly and update the paper.

**Post rebuttal**
Rebuttal read and acknowledged, score updated

---

> ### Author Response · Authors · 2022-08-02
> **Response to Reviewer 3fFh**
>
> Thank you for dedicating your time and energy to reviewing our paper. We would like to provide point-to-point responses as follows.
>
> **Q1. Unlabeled Graph**. Both unlabeled and labeled graphs are handled based on the classifier (fully-connected layers by default) in our framework for graph label prediction. With this classifier, we can keep the prediction of augmented representation the same as the prediction of original representation for unlabeled graphs. On the other hand, for labeled graphs, we just keep the prediction of augmented representation the same as ground-truth labels. In this way, we achieve label-invariant for both unlabeled and labeled graphs. During training, the improvement of the classifier would help with the label-invariant augmentation for both unlabeled and labeled graphs, which in turn benefits the classification performance. To clarify how we handle the unlabeled graphs, we will add descriptions in the "Label-invariant Augmentation" part of Section 3.2 in revision as follows:
>
> *Based on the classifier formulated in Eq. (3), we define label-invariant as follows. For labeled graphs, label-invariant means the predictions of augmented representations by the classifier are the same as their corresponding ground-truth labels. For unlabeled graphs, label-invariant denotes that the predictions of augmented representations and the predictions of original representations by the classifier are the same.*
>
> **Q2. Meaning of "Closest to The Decision Boundary."** The candidate that is closest to the decision boundary of the classifier meets two requirements. The first one is label-invariant (as illustrated in Q1 for unlabeled and labeled graphs), and the second one is having the least probability of belonging to the same class as original representation/ground-truth label for unlabeled/labeled graph among all qualified candidates based on the classifier (as described in the caption of Fig. 2). We add more interpretation on this in the "Label-invariant Augmentation" part of Section 3.2 in revision as follows:
>
> *To achieve the label-invariant augmentation, for each graph, we randomly generate multiple perturbations and select the qualified augmentation candidates that obey the label-invariant property. Among these qualified candidates, we choose the most difficult one, i.e., the one that has the least probability of belonging to the same class as ground-truth label/original representation for labeled/unlabeled graph, to increase the model generalization ability.*
>
> **Q3. Description of "Hard Positive Mining."** We agree that we choose a lightweight technique by picking a hard example within a sampled set. As our paper does not focus on proposing an approximate bilevel optimization scheme, we will remove this sentence in the contribution part of Section 1 in revision as follows:
>
> *In the rich representation space, we aim to generate the most difficult sample for the model and increase the model generalization. We choose a lightweight technique by randomly generating a set of qualified candidates and selecting the most difficult one, i.e., minimizing the maximum loss or worst case loss over the augmented candidates.*

---

> > ### Comment · Reviewer_3fFh · 2022-08-06
> > **Acknowledgement of rebuttal**
> >
> > I have read the response to my review and the very thorough responses to the other reviews. The authors improved the clarity of the paper and fixed the last issues. Score updated.

---

> > > ### Author Response · Authors · 2022-08-06
> > > **Response to Reviewer 3fFh**
> > >
> > > Thank you very much for reviewing our paper and prompt response. We are happy to address all the concerns and feel appreciative of the increased score.

---

### Official Review · Reviewer_e8z3 · 2022-07-11

**Rating:** 6
**Confidence:** 4
**Soundness:** 3 good
**Presentation:** 3 good
**Contribution:** 3 good

**Summary:**

This paper presents a label-invariant augmentation for graph-structured data. It conducts the augmentation in the representation space and augments the most difficult sample while keeping the label of the augmented sample the same as the original sample. Experimental results on eight benchmark graph-structured data show that the developed metho outperforms classical GNN-based methods and recent graph contrastive learning.

**Questions:**

a. The model sizes and inference time of different methods in Table 2 are not discussed.
b. In Figure 2, it is unclear whether the number of augmented representations affect the final result.


**Limitations:**

The authors have discussed the limitation and potential negative societal impact.

**Strengths And Weaknesses:**

Strengths:
a. This work presents a label-invariant augmentation strategy for graph contrastive learning by involving labels in the downstream task to guide the contrastive augmentation.
b. The authors present a lightweight technique to randomly generate a set of qualified candidates and selecting the most difficult one for increasing the model generalization.
c. Experimental results on eight graph benchmark datasets have verified the effectiveness of the developed label-invariant augmentation.
Weaknesses:
a. It is unclear why selecting the hardest augmented sample can enhance the performance. Why selecting top-K hardest augmented samples?
b. Failure cases of the develop method are not discussed.
c. As shown in Table 2, the results of the developed method are not always best among different datasets. Please explain such results.

---

> ### Author Response · Authors · 2022-08-02
> **Response to Reviewer e8z3**
>
> Thank you for dedicating your time and energy to reviewing our paper. We would like to provide point-to-point responses as follows.
>
> **Q1. The Hardest Augmented Sample**. The hardest augmented sample tends to be more close to the decision boundary compared to a random sample and the easiest sample. As the classifier is learning to get a better decision boundary to separate samples from different classes, a harder sample would contribute more to the classifier training, thus enhancing the performance. Figure 4(d) shows the performance of different augmentation strategies among qualified candidates, including the most difficult augmentation, random augmentation, and the easiest augmentation. We can see that the most difficult augmentation increases the model generalization and indeed brings in significant improvements over the other two ways.
>
> **Q2. Why Selecting Top-K hardest augmented samples**. Sorry for the confusion. As other contrastive learning works, we also choose one sample for augmentation. Specifically, we choose **one** hardest sample from 10 candidates that satisfy the label-invariant constraint for augmentation.
>
> **Q3. Failure Cases and Not The Best in Table 2**. Thanks for this good suggestion! We believe the failure cases come from the violations of the label-invariant constraint. Although we explicitly add the label-invariant constraint during our augmentation, our constraint relies on the learned classifier, which can be a proxy of ground truth. Therefore, there still exist some violations of our method (See Figure 3(b)). Our method does not achieve the best performance on *COLLAB*, *RDT-B*, and *GITHUB*, where our method suffers from a relatively lower label-invariant rate than other datasets.
>
> **Q4. Mode Size and Execution Time**. For model sizes, we take the model size of GraphCL as a baseline, and then evaluate others. Here GAE, SimGRACE, GraphCL, and GLA are adopting the same number of model parameters with different losses or augmentation strategies, therefore their model sizes are the same. For execution time, we report the pre-training and fine-tuning time of compared methods and training time of our method. All experiments are conducted on a physical machine with Ubuntu 18.04, total memory of 64GB, an AMD Ryzen Threadripper 2920X 12-Core Processor, and an NVIDIA GP102 GPU. Note that due to GPU memory issue, experiments of MVGRL on *RDT-B*, *RDT-5K*, and *GITHUB* only used CPU, which results in a relatively high time cost. Overall, SimGRACE is the fastest, and our method is the second fastest, and both are significantly faster than other methods. We believe this is because SimGRACE and our method do not need to generate augmented graph data.
>
> Table Q4: Mode sizes (based on GraphCL) and execution time (s).
> | Method | Model Size | *MUTAG* | *PROTEINS* | *DD* | *NCI1* | *COLLAB* | *RED-B* | *RED-5K* | *GITHUB* |
> |---|:---:|:---:|:---:|:---:|:---:|:---:|:---:|:---:|:---:|
> | GAE | 1.00 | 24 | 153 | 1390 | 517 | 1150 | 2812 | 6662 | 2475 |
> | Infomax | 1.28 | 19 | 210 | 1545 | 614 | 1931 | 3886 | 11388 | 6865 |
> | MVGRL | 1.69 | 34 | 412 | 1993 | 631 | 2755 | 220787 | 313019 | 76246 |
> | SimGRACE | 1.00 | 7 | 31 | 74 | 106 | 474 | 197 | 529 | 948 |
> | GraphCL | 1.00 | 14 | 104 | 1270 | 489 | 1086 | 1847 | 2600 | 1697 |
> | JOAOv2 | 2.11 | 2740 | 2954 | 5088 | 3539 | 12502 | 7963 | 17087 | 11094 |
> | GLA (Ours) | 1.00 | 31 | 65 | 183 | 195 | 859 | 224 | 617 | 879 |
>
> **Q5. Number of Augmented Representations**. For the number of augmented representation candidates, we believe there is no significant difference when the number is small. For the number of selected augmented representations, we only choose to have one hardest sample in the current version. In the following table, we show the results of our method with top 1, top 2, top 3, and top 10 selected samples. As the number of candidates is set to 10, the top 10 works as a randomly selecting strategy illustrated in Fig. 4(d). From this table, we can see that when the number of selected augmented representations is small, the performance of GLA does not change much. However, as this number increases, the augmentation strategy is more like randomly selecting than choosing the hardest sample, thus may hurt the performance in most cases, which is theoretically supported by the recent study [1].
>
> Table Q5: Semi-supervised graph classification results of our method with different numbers of augmented representations under 50\% label rate.
> | Method | *MUTAG* | *PROTEINS* | *DD* | *NCI1* |
> |---|:---:|:---:|:---:|:---:|
> | Top 1 augmented representation (Ours) | 90.00 | 76.19 | 80.22 | 80.66 |
> | Top 2 augmented representation | 89.42 | 76.19 | 80.30 | 80.26 |
> | Top 3 augmented representation | 89.95 | 76.27 | 80.14 | 80.77 |
> | Top 10 augmented representation (Random) | 88.95 | 76.43 | 78.64 | 79.56 |
>
> >[1] Gong, Chengyue, et al. "Maxup: Lightweight adversarial training with data augmentation improves neural network training." *CVPR*. 2021.

---

### Meta-Review · Area_Chair_sM4g · 2022-08-27

**Recommendation:** Accept
**Confidence:** Certain

**Metareview:**

The paper finally received three unanimous scores 6/6/7 and all the reviewers think the authors have addressed their initial concerns. The AC also think positive to this work and suggst to accept this paper.

**Award:**

No

---

### Decision · Program_Chairs · 2022-09-14

Accept